# Graduate Students, Community Partner, and Faculty Reflect on Critical Community Engaged Scholarship and Gender Based Violence

**Mavis Morton [1,\*], Annie Simpson [1], Carleigh Smith [1], Ann Westbere [2], Ekaterina Pogrebtsova [3] and Marlene Ham [4]**

[1]  Department of Sociology and Anthropology, University of Guelph, 50 Stone Rd. E., Guelph, ON N1G 2W1, Canada; asimps13@uoguelph.ca (A.S.); csmith63@uoguelph.ca (C.S.)

[2]  School of English and Theatre Studies, University of Guelph, 50 Stone Road E., Guelph, ON N1G 2W1, Canada; westbera@uoguelph.ca

[3]  Department of Psychology, University of Guelph, 50 Stone Road E., Guelph, ON N1G 2W1, Canada; epogrebt@uoguelph.ca

[4]  Ontario Association of Transition and Interval Houses, PO Box 27585 Yorkdale Mall, Toronto, ON M6A 3B8, Canada; marlene@oaith.ca

\*  Correspondence: mavis.morton@uoguelph.ca

**Abstract:** This article reflects on the challenges and opportunities associated with community engaged learning at the graduate level, and challenges higher education to do more to support the teaching–research–service nexus. The community university partnership involved a graduate student class, a faculty member, and a community member from a provincial not for profit association. We examined our principled and collaborative process of critical community engaged scholarship geared toward addressing violence against women, and more specifically, femicide. Our research resulted in knowledge mobilization tools that could be used to inform various audiences (e.g., women's shelter staff, the public, government, and journalists) about how mainstream media sources report and portray the issue of femicide. Our work had an explicit social justice focus with aims to generate a better understanding of the structural causes of violence against women and historically-created gendered hierarchy and its ongoing impacts. This paper offers insights for others interested in pursuing community engaged research within a community engaged learning environment.

**Keywords:** critical community engaged scholarship; community engaged research; community university partnerships; knowledge mobilization; femicide; gender-based violence; social justice; graduate student; community partner; faculty; reflection; experiential education

## 1. Introduction

This article, collaboratively written by graduate students, a faculty member, and a community partner, outlines our collective experiences conducting community engaged scholarship (CES) within a graduate sociology course at the University of Guelph. Our CES project falls under the category of critical community engaged scholarship (CCES) (Gordon da Cruz 2017), which seeks to address public issues by drawing attention to their structural causes. According to Gordon da Cruz (2017), this type of collaborative research supports university and community partnerships in producing knowledge that dismantles systemic sources of racial and social injustice. As a community university (CU) partnership, we collaboratively conducted research to co-create and mobilize knowledge about the prevalence and seriousness of violence against women in our society in an effort to work toward gender equality and

social justice. Specifically, our work shed light on femicide[1] as the most extreme form of violence against women and girls (Corradi et al. 2016). In what follows, we position our work within the CES, CCES, and community engaged learning (CEL) literature to demonstrate how this aligns with work on community wellbeing (Hall 2009) and resilience from a social justice perspective. We provide a summary of our project within a graduate sociology course and highlight common benefits and challenges of CES and CEL from student, faculty, and community partner perspectives. We hope that these perspectives will assist others in their efforts to work toward social justice by employing a critical community engaged research framework within higher education.

## 2. Community Engaged Learning and Community Engaged Research in Higher Education

There is widespread agreement that the challenge for higher education is to engage with society in an integral way that improves research, teaching and learning, and collaborates in social transformation (Cortez Ruiz 2014). The nomenclature to describe such engagement and literature about how to collaborate has received much attention within higher education (Barker 2004; Wenger et al. 2012; Bateman 2018; Kantamneni et al. 2019) and outside of it (Barge et al. 2008). This attention increased with the publication of Boyer's (1996) foundational discourse, in which he called upon colleges and universities to " . . . become a more vigorous partner in the search for answers to our most pressing social, civic, economic, and moral problems" which he referred to as the "scholarship of engagement" (p. 18). Based on a scoping review of the literature over the past 20 years, Beaulieu et al. (2018) conceptualize "engaged scholarship" as building " . . . mutually beneficial and reciprocal bridges between university activity and civil society" in service to society rooted in values of social justice and citizenship (p. 13). There are many ways to refer to and integrate teaching–research–service that can align with the philosophy of engaged scholarship. For example, frameworks such as Action Research (AR) (Chandler and Torbert 2003; Greenwood 2007), Community Based Research (CBR) (Strand 2003; Halseth et al. 2016), Community Based Participatory Research (CBPR) (Coughlin et al. 2017; Tremblay 2009; Minkler and Wallerstein 2008), Service Learning (SL) (Levkoe and Stack-Cutler 2018; Stanton 2000), Community Service Learning (CSL) (Taylor and Kahlke 2017), and Community Engaged Scholarship (CES) (Gelmon et al. 2013a; Gordon da Cruz 2018) are some of the most common. The framework we used is Community Engaged Scholarship. While many of the above approaches have similarities, we focus on CES because the purpose of the course was to introduce graduate students to the principles and practices of CES specifically. As articulated by Barker, "[T]he language of engagement suggests an element of reciprocal and collaborative knowledge production that is unique to these forms of scholarship" (Barker 2004, p. 126).

There is now an abundance of academic literature that identifies common principles and practices associated with ethical and sustainable CES (see for example: Beaulieu et al. 2018; Baker et al. 1999; Community-Campus Partnerships for Health 2013; Davis et al. 2017). Of these, some of the most common principles include:

- Mutually beneficial relationships that are founded on respect, trust, genuine commitment, and shared goals;
- Community identified need that works toward social justice and the betterment of the community;
- Collaboration that is reciprocal, values mutual learning, recognizes multiple assets of partners, and values a multi-disciplinary approach;
- Communication that is clear, honest, and respectful in which listening is paramount;
- Balancing power which means shared input, decision making, and sharing resources;

---

[1] Femicide is commonly defined as the intentional murder of women by a man because they are women (Garcia-Moreno et al. 2012; Ferrara et al. 2015).

- Meaningful outcomes, which requires tangible and relevant goals that focus on social justice and that promotes diversity, academic, and community facing results.

Common processes and practices associated with CES include the following (see for example: Janzen et al. 2017; Williamson et al. 2016; Watson-Thompson 2015; Garner 2015):

1. Identify and develop relationships;
2. Establish partnership engagement goals;
3. Engage in ongoing, collaborative partnership activities;
4. Conduct critical scholarly investigation of public issues;
5. Assess outcome and impact.

Despite the common principles and practices, there are various approaches to community engagement. Such variations are recognized under the larger construct of CES, which encompasses "... intellectual and creative activities that generate, validate, synthesize, and apply knowledge through partnerships with people and organizations outside of the academy" (Seifer 2012). Although CES can encompass activities other than "research", we conducted research and mobilized our findings within a CES framework. Thus, we refer to our work more specifically as community engaged research (CEnR) to highlight the research process that was central to developing our knowledge mobilization tool. Our decision to use the CEnR term is further explained in the student reflections below. CEnR has been defined as:

> ... a collaborative approach to research that democratically involves community participants and researchers in one or more phases of the research process. Partners share responsibilities and leverage their unique strengths to enhance understanding of the target of research (often a social or cultural dynamic of the community) and integrate the derived knowledge with action to improve the well-being of community members. (Nation et al. 2011, p. 90)

While our work falls under the umbrella term of CES, and more specifically, CEnR, it can also be appropriately referred to as community engaged learning (CEL), as it occurred within a graduate course. CEL is a teaching and learning pedagogy that meaningfully integrates community engagement and curricular programming with intentional alignment between course learning outcomes and community identified needs (Morton et al. 2018). CEL involves mutually beneficial collaboration for the purposes of co-learning and co-creating relevant scholarship or scholarly activity that strengthens academic inquiry, personal and professional development, and contributes to positive social change/justice (this draws on definitions from: Carnegie Foundation for the Advancement of Teaching 2017; Boland 2013; Bringle and Hatcher 1995; Gordon da Cruz 2017; Kleinhesselink et al. 2015; Johnson-Curiskis and Wolter 2004; Morton 2013; Weigert 1998). We intentionally adopt the term "community engaged learning" (Delano-Oriaran et al. 2015) to reflect the centrality of the community in both the pedagogy and the research (Brudney and Russell 2016, p. 278). Thus, the emphasis on community engagement is an important feature that distinguishes CEL between comparable approaches and nomenclatures. Our focus on CEL and CES is not meant to exclude researchers accustomed to other related approaches, but rather increase interest in, and understanding of, the community engagement principles and practices that we encourage scholars to apply to their work.

There is limited academic literature on CEL. However, the teaching–research–service nexus is not new and reflections and research focusing on different ways of addressing this continuum is expanding. For example, there is an abundance of literature on service learning (see Ash et al. 2005; Levkoe and Stack-Cutler 2018; Jagla et al. 2015) and similar approaches referred to as "Community Based Service Learning", and "Community Service Learning" (Taylor et al. 2015) which share some similarities as well as differences with CEL. One such difference is that service learning emphasizes a "service" component that students provide outside of the classroom with the associated critical reflection about how their service connects to the course. CEL, on the other hand, refers to activities between students, faculty, and community partners that respond to a community identified need and

result in co-created disseminated knowledge/products. A similarity between the two is the reflective processes that often make up part of students' curricular experience and assessment (Ash et al. 2005). In fact, reflexivity is an important part of any CES (Levkoe et al. 2014; Janzen et al. 2017; D'Enbeau et al. 2013). It can be used as an evaluative tool to assess the extent to which the CU partnership accomplished its shared goals and lived the principles of CES necessary for ethical, successful, and sustainable CES (Sadler et al. 2012; Janzen et al. 2017). Course-embedded reflection is also common to assessment in higher education and a useful way to assess student learning outcomes related to knowledge, skills, and values (Holland et al. 2008). Ash and Clayton developed a reflection framework that includes paragraphs written by students reflecting on a series of questions requiring them to describe and analyze their experiences of service learning (Ash et al. 2005). In the Methods section below, we explain the structured critical reflection (Levkoe et al. 2018; Brandenburg and Wilson 2013) assignment that became the foundation for this paper.

As the faculty member and community engaged scholar, the experiences and reflections from the students and community partners that make up the partnership are invaluable to me. In keeping with the principles of CES and other models including Action Research (Salm 2013) and Community Based Participatory Research (CBPR) (Tremblay 2009; Janzen et al. 2017), I have a responsibility to ensure the needs of the community partner are addressed to the best of our ability and in line with our common objective. As the course instructor, and as a CES mentor for graduate students, I have a responsibility to provide a teaching and learning environment that assists students' development, and demonstrates course, program, and university level learning outcomes. Therefore, the practice of communicating with and listening to all partners is an important mechanism for continual improvement to live the principles of CES, while also employing best practices in teaching and learning.

## 3. Community University Partnership Case Study Overview

In the winter semester of 2017, a graduate sociology class at the University of Guelph partnered with the Ontario Association of Interval and Transition Houses (OAITH) and the College of the North Atlantic on a research project pertaining to femicide, commonly defined as the intentional murder of women by a man because they are women (Garcia-Moreno et al. 2012; Ferrara et al. 2015). As a community engaged scholar, the sociology professor recently designed this course to offer graduate students from sociology, as well as any other disciplines within our institution, an opportunity to learn about and practice CES. Graduate training and mentorship on CES is lacking within institutions of higher education (Levkoe et al. 2014) despite research that identifies the potential benefits for undergraduate and graduate students (Levkoe et al. 2014; Gelmon et al. 2013b; Dinour et al. 2018). The associated learning outcomes for this course included: understanding CES, including the underlying principles that support its ethical and sustainable practice, and the development of skills and values by applying this knowledge to work collaboratively on a social problem identified by a community partner. In this case, the community partner knew the faculty member prior to the start of this course.

OAITH and the faculty member developed a mutually beneficial community campus partnership over several years to work on projects related to violence against women in Ontario. The faculty member and undergraduate students work with this provincial association to track and record information for OAITH's annual Femicide List (elaborated below), which they use as part of their education and advocacy to address the prevalence and seriousness of violence against women in Ontario. An important part of OAITH's mission is to " . . . advocate for systemic change in order to end violence against women and their children, through advocacy, collaboration, and education" (OAITH). As such, the femicide work, which OAITH has been doing since 1995, is a good example of the ways in which they work toward their mission.

The faculty member and community partner identified this partnership and engaged research project in the hope that it would be a realistic and appropriate endeavor for students given the time, skills, and resources available over a 12-week term. Our partnership included six graduate students from various disciplines (including three Ph.D. students from geography, psychology, and sociology,

and three MA students from sociology and a criminology criminal justice policy program), a sociology professor whose research focuses on violence against women, as well as community engaged learning (CEL), and the community partner, who has extensive knowledge and experience pertaining to violence against women.

In line with the principles of CES (Beaulieu et al. 2018; Gordon da Cruz 2018; McNall et al. 2009; Northmore and Hart 2011; Williamson et al. 2016) this project responded to a need identified by OAITH at the outset. For over 20 years, OAITH has utilized mainstream media sources to identify and consolidate the names, pictures, and short biographies of victims, which is then added to their femicide list. OAITH relies on this list to educate the public on the topic of violence against women, and for their associated advocacy work[2]. Through ongoing conversations with the faculty member, OAITH expressed an interest in expanding the Ontario femicide list beyond the descriptive data. Specifically, they expressed the need to critically examine the ways in which femicides are portrayed throughout various mainstream newspaper reports, and to determine how this can impact the ways in which femicide is understood and subsequently addressed by the public, the state, and public policy.

Accordingly, the main goal of the CES project was to create a knowledge mobilization tool that OAITH could use to inform and educate different audiences (e.g., Ontario women shelter staff, the public, government, journalists, etc.) about how Ontario media sources report and portray the issue of femicide. By the end of the course, we achieved this output. See https://www.oaith.ca/oaith-work/current-initiatives.html for the final knowledge mobilization products co-created by this CU partnership.

To achieve the project goal, we first conducted a scoping review of the literature pertaining to media portrayals of femicide. Using the results of this literature review, we created a framing analysis[3], which was then used to critically analyze news reports of the 20 victims of femicide in Ontario between 2015 and 2016. This analysis examined the reporting of femicides across three types of news outlets (i.e., national newspapers, local newspapers, and national TV news sources). The methodology used to construct the framing analysis tool reflected previous research on media framing of violence against women (Easteal et al. 2015; Richards et al. 2014; Comas-d'Argemir 2015; Gillespie et al. 2013).

This CES project is a way to work toward gender equality and social justice given the impact and influence mainstream media can have on the way a social problem is understood and addressed (Major 2015). The ways in which the news media frame violence against women and femicide influences how society understands such violence, as well as solutions and public responsibility (Gillespie et al. 2013; Comas-d'Argemir 2015; Taylor 2009). Further, research conducted by Wright and Washington (2018) highlights the significant role media plays in shaping the public's awareness of, imagination about, and attitude toward crime and crime victims. Specifically, we know from previous research that femicide victims, as well as survivors of gendered violence, tend to be portrayed in mainstream media in ways that victim blame (Corradi et al. 2016). Further, such portrayals generally fail to acknowledge

---

[2]　For example, once the Femicide List for the current year has been created, it is distributed throughout Ontario through its membership and beyond. Local shelters can use the list as part of their annual December 6th vigils held to commemorate the Montreal Massacre in 1989, whereby 14 women were murdered because they were women on the last day of classes at Montreal's Ecole Polytechnique.

[3]　The Media Framing Analysis was a multi-step process. The first step involved conducting a scoping review of the research surrounding media portrayals of femicide cases. We identified studies using a combination of terms (e.g., "femicide", "media", and "representation"). We then uploaded the identified studies to Mendeley, a reference management software, which was further sorted for relevance. The next step was to summarize each study using a Google Document table that captured the purpose, methodologies, and findings of each article. We used the summary table and the original academic articles to write individual literature reviews in order to extract the common frames identified in other research about the way femicide and violence against women is represented within mainstream media. We then compared and contrasted the identified frames to determine commonalities and areas of divergence. This included four positive frames (victim humanized, femicide label, picture of victim, and gendered social problem) and four negative frames (victim blaming, voice of authority, individualized, and history of violence against women undocumented). Following this synthesis, the team utilized these eight frames as codes to analyze current media portrayals of femicide. We examined a total of 29 Ontario femicide cases in 2016–2017 by studying 73 news items from three media sources (i.e., mainstream national newspapers, local newspapers, and TV news).

the history of male violence that preceded the murder, and individualize the problem in ways that are not connected to or understood as part of a larger problem of gender-based violence, patriarchy, and gender inequality (Corradi et al. 2016; Comas-d'Argemir 2015).

Accordingly, members of this partnership recognized the need for research on the way in which Ontario femicide victims are represented in mainstream local and national press, and to disseminate these findings in a way that engages multiple audiences to function as an agent of positive social change. We deliberately designed our knowledge mobilization tool to advance the province of Ontario's understanding of femicide as a gender-based social problem and move forward on gender equality initiatives, thereby increasing social justice (Poortinga 2012; McCrea et al. 2014, 2016). Although it is still too early to assess the full impact of our knowledge mobilization outputs, our community partner has provided us with feedback and metrics that help to exemplify the immediate outcomes of our work. Specifically, our online femicide visualization has had over 1700 views which continue to increase steadily every year, indicating substantial public reach. Moreover, since its online publication, our partner has been using our femicide research and knowledge mobilization tool in conversations with their funders and journalists. Our partner has credited our research and tool for helping them present a stronger case for increased funding for femicide resources, training, and public awareness in the last two recent years.

## 4. Understanding Femicide as a Gender-Based Social Problem

Our CEnR on femicide is relevant to the research on community well-being and resilience. Current research in this area conceptualizes resilience as " ... the capacity to intentionally mobilize [ ... ] people and resources to respond to, and influence social and economic change ... " (Colussi 2000, pp. 1–5). Other definitions and models of community resilience often include a component of social solidarity, in which everyone feels respected and heard. For this to be realized, greater social equality among all members (irrespective of gender, age, or race) is required. Working toward equity requires an understanding and acknowledgement of the structural causes of violence against women and the historically created gendered hierarchy (Darychuk and Jackson 2015). Gendered violence, intimate partner violence, and femicide are inseparable from these contexts from which they derive. As such, approaches to ending femicide must incorporate structural and ideological changes and not solely rely on solutions (i.e., services) aimed at the individual and/or family (Burnette and Hefflinger 2017). A critical feminist conceptualization of this issue means working to challenge explanations of gendered violence that hold women responsible for their own victimization (Wright and Washington 2018; Easteal et al. 2015) and one way to do this is to change the way mainstream media portray and report femicide.

As mentioned above, we characterize our work on this issue as critical CES. We adopt the language of critical CES to highlight our intention to work on issues in a way that illuminates the structural causes and attempts to uncover effective solutions (Gordon da Cruz 2017, p. 368). Gordon da Cruz also uses the word "critical" to refer to a justice-oriented practice of engaging with communities so that scholarship is more practically relevant and intentionally works with marginalized communities to impact positive change (Gordon da Cruz 2017, p. 368). In these ways, we acknowledge and identify the "critical perspectives and commitments to justice" that motivate our work (Gordon da Cruz 2017, p. 370). Our commitment to social and racial justice, and interrogating the structures that maintain inequality, aligns with our community partner's justice aims in theory and practice. OAITH's mission identifies their objectives as " ... advocat[ing] for systemic change to end violence against women and their children, through advocacy, collaboration, and education. OAITH is committed to operating within a feminist, anti-racist, and anti-oppression, intersectional framework" (OAITH 2018).

In addition, we embraced studying social issues from a sociological perspective, referred to as using a "sociological imagination" to study social problems. Sociologist C. Wright Mills used the term to suggest that students of sociology should understand and acknowledge the historical, cultural, environmental, and social processes that directly and indirectly cause social problems in contemporary

society (e.g., poverty, racism, and sexism) (Hironimus-Wendt and Wallace 2009, p. 77). Mills believed that the primary lesson of sociology should be that " … humans have the potential to reduce or resolve most social problems" (Hironimus-Wendt and Wallace 2009, p. 78). This perspective incorporated a pragmatic role for the discipline, requiring researchers to use their sociological imagination to make a positive difference in our own lives and in our communities. This disciplinary focus fit well with our and our community partner's intention to work on femicide and violence against women from a critical feminist perspective (Corradi et al. 2016).

## 5. Student, Faculty, and Community Partner Reflections on Critical Community Engaged Scholarship

Although we came together in a course format, we refer to our collaborative work as CEnR, which is a type of scholarly activity that commonly occurs within a CES framework and involves a collaborative research process to produce shared knowledge. We (the students) attest to the fact that this research extended beyond a university course and became a research project, a community campus partnership, and a commitment to work toward community well-being and positive social change. We saw ourselves as more than students in a traditional graduate course, and rather, as researchers and emerging community engaged scholars working in collaboration with a community partner on the relevant social and political issue of femicide. In the following section, we discuss reuniting as a research team for this paper, and outline the process of determining which content to include from our individual reflections, as well as how to organize and present this information.

## 6. Methods

A component of the graduate course, entitled "Principles and Practices of Community Engaged Scholarship", included a week of readings on the topic of "Scholarship and Assessing CES". The academic literature in this area identified some of the challenges that faculty and emerging scholars face, including finding a balance between the competing demands of producing both outputs that are of value/direct use to the community, and more traditional academic outputs (Calleson et al. 2005; Fitzgerald and Primavera 2013; Cavallaro 2016; Gelmon et al. 2013b). After reading about these challenges, what emerged from a class discussion was an interest in, if not a commitment to, writing an academic journal article about our unique experience of this CU partnership and our CCES. Almost a year and a half after the course ended, we were inspired to carry out what we had originally hoped to do.

A common pedagogical teaching and learning activity and assessment practice in CEL and service learning courses include critical reflection exercises as a vehicle for learning (Levkoe et al. 2014, p. 70). As part of this course, 10% of the student evaluation came from a reflection dossier, which included three written reflections from each student " … in week 3, 8, and 12 to encourage [students] to think about and capture [their] learning (personal and academic development) about community engaged scholarship, [their] community engagement activities, experiences, and other course learning outcomes" (Morton 2017). In so doing, we were asked to respond to specific questions posed by the instructor, which often required us to reflect on the ways in which our experience of CEnR and CES aligned or did not align with other students, faculty, and community partners based on course readings. The information outlined in the section below elaborates salient passages from these reflections.

During the summer of 2018, one of the students from the course emailed the instructor asking if she was interested in writing an article for this special edition given what he thought was a good fit. The instructor contacted all the students via email, and with the exception of one student who was no longer available to participate, the research team expressed a unanimous interest in pursuing this opportunity. Over the course of the summer, we met in person to discuss how we would initiate the writing process. Shortly thereafter, we transitioned to communicating online through a Google Document, which allowed us to communicate more frequently and in real time from our various locations. During these initial meetings, we referenced our independent student reflection dossiers

to establish larger, common themes amongst the group. Once identified, we added these themes to the Google Document, and continued to add relevant information under each theme. This was an inclusive way to provide all authors with the opportunity to participate in what turned out to be another interesting collaborative writing endeavor. The faculty member agreed to write a section that came from reflecting on her experience of the CU partnership and this was also informed by the student and community partner reflections.

The community partner reflection came from an interview conducted with the community partner following the completion of the course. This interview was conducted by our community engaged learning coordinator, who works for a CES brokering unit called the Community Engaged Scholarship Institute (CESI) at the University of Guelph. The community partner was a research participant involved in the scholarship of teaching and learning (SoTL) research on community engaged learning that had been conducted by two sociology faculty (including the faculty who is one of the authors of this paper) and the community engaged learning coordinator mentioned above. This graduate course served as one of the research sites, and as part of this research, the researchers wanted to hear from all partners in the CU partnership. With the community partner's permission, we have the benefit of including her reflections on our partnership and project.

## 7. Student Reflections on Community Engaged Scholarship and Community Engaged Research

In this section, we reflect on the common challenges and opportunities that came from our experience as graduate students, a community partner, and as a faculty member learning about, and trying to employ, the principles of critical community engaged scholarship within a graduate sociology course to address a community identified need.

Our intention in this paper was to elaborate our common challenges and identify where we experienced opportunities. In reflecting on this process, we recognized that it was most appropriate to address the challenges and benefits under larger themes, as they were often interrelated. The reflections below address similar issues and tensions identified by previous research on community engaged scholarship and community engaged learning, which we read throughout this course.

### 7.1. Iterative Process and Uncertain Outcomes

As is common in CES, CBPR, PAR, and AR, the process of understanding, clarifying, and scoping our deliverables for this project was ongoing and iterative (Janzen et al. 2017; Morton 2013). Initially, the purpose of our research and the subsequent deliverables were not clearly defined. Rather, our community partner communicated a general interest in critically examining Ontario femicide data. Over the course of the term and with ongoing communication with the community partner, we agreed to work on two unrelated pieces of research pertaining to femicides in Ontario (see Morton et al. 2017, p. 5). Although our research questions and methodologies became clearer over time, the exact outputs that would go to the community partner remained elusive throughout much of the term. Ultimately, the iterative nature of this process posed common challenges and benefits for the student researchers.

Members of the team predominantly identified the challenges of flexible research at the onset of the research process. For example, one student stated: "In the beginning, it was hard for our group to see what the end-goal of our project would be because we had no prior experience working within a process where there were not fully constructed or foreseeable outcomes". Upon further consideration it is likely that this challenge was exacerbated by the fact that many of our academic outputs had, until this point, taken the form of traditional written papers, and most of us had little to no experience producing alternative outputs. As such, we initially lacked the skills to adapt to the changing targets and engage in ongoing negotiation with the partner to ensure their needs would be met. This was a steep but important learning curve for us, as we were reminded that this process is more reflective of the conditions in the working world.

In the latter half of the course, our team identified two benefits of conducting iterative research. First, through consistent communication and negotiation, we developed and modified a series of

deliverables that transitioned alongside and best reflected the developing needs of our community partner (Tarantino 2017). As one student stated, this afforded us the opportunity to "continuously gain feedback and new directions based on changing real-word needs". Further, this allowed us to utilize our time and resources in the most appropriate and effective manner.

Second, we recognized that iterative research reflects real world situations, rather than the often-isolated process of writing a paper (McLaughlin 2010; Hironimus-Wendt and Wallace 2009). One student summarized this process, stating: "At first, it was difficult for me to adjust to the flexible nature of this process, as it contrasts with the rigid process that is foundational to traditional research. However, I have come to appreciate that CES mirrors real-world employment situations rather than highly structured academic arenas". Another student elaborated this perceived benefit, stating: "I believe that adaptability is a skill that is fundamental for all our future careers (whether in academia or in the "real world") but often under-looked and even thwarted in many classic university courses".

Towards the end of the course, some members of our team experienced challenges around sharing the responsibility for knowledge mobilization, and the ownership of knowledge mobilization (KMb) products, with our community partner. As one student asked: "Where do my responsibilities end? Do I hand over the product to our partner, and let them exercise their best judgement about how to best mobilize it?" We addressed this challenge by recognizing that we can continue to support our community partner in their knowledge dissemination and advocacy efforts if "our products are accompanied with a 'transmittal' that describes our expectations and outlines recommendations for how our KMb products might best be used".

### 7.2. Consistent Communication

Due to the flexible nature of CES work, designing a project is a negotiation process that develops over time, and requires constant communication and modification. Specifically, CES relies on a relationship between the researchers and the community partners that includes both trust and genuine commitment. Further, it requires clear communication between these parties to design a project that is both meaningful and mutually beneficial. Our team identified benefits related to the communication aspect of this research, including improved communication skills with both team members and with our community partner/future partners.

With respect to the former, our team engaged in consistent direct communication (weekly in-class meetings), as well as indirect communication (email and online group discussions). This provided us with greater opportunities to brainstorm, work through ideas, and learn throughout all stages of the research. We were further encouraged to express our ideas in a clear and concise manner that was conducive to diverse learning styles, and to address and manage opposing views in a respectful manner. Moreover, one student suggested that our classroom discussions "offered an opportunity for formative learning".

To ensure the negotiation aspect of our partnership, we aimed to communicate with our community partners in real time, and on a regular basis using online tools such as Google Documents and Skype, as well as frequent phone calls and emails. This allowed us to continuously gain feedback regarding our outputs and address outstanding questions. For example, one student highlighted how such interactions provided clarity, stating: "In our most recent class, we had a video call with our community partner from OAITH. This was extremely helpful as we were able to ask questions and find out more about OAITH's vision".

### 7.3. Working as a Team

As the majority of us primarily work independently on our research, members of the team commonly discussed the benefits and challenges of working with others. With respect to the benefits, working as a team allowed us to build on each other's ideas, and to collaboratively explore the best ways to communicate our findings. One of the most memorable examples of this occurred when we were designing the visual format of our knowledge mobilization tool. Initially, we wanted

our work to highlight the negative and positive frames utilized by the media when reporting cases of femicide. To help with this process, we invited the Knowledge Mobilization Coordinator from the Community Engaged Scholarship Institute (CESI) to our class to do a presentation on what knowledge mobilization is and to identify different ways to communicate research findings based on the audience and community partner needs. We started our creative process by drawing a "poster" on the chalkboard, and tentatively wrote "positive frames" and "negative frames" on opposing sides. One student suggested that we use different colors (green and red) to differentiate between the positive and negative frames, respectively. From there, a student suggested that we call this a "Heat Map" as we were essentially representing the "hot spots" of these media reports. Ultimately, this collaborative process allowed us to produce an effective visual element to communicate our research findings in an engaging and interactive way.

Moreover, some students enjoyed the social aspect of this research, whereby "certain challenges that come along with research, such as what can commonly be an isolating experience of conducting rigorous scholarly work, felt entirely different with our class structure and CES framework" (student reflection). This type of co-creative work provided opportunities to build on each other's knowledge and skills.

That said, collaboration was not always easy, and we experienced a number of challenges. In particular, our team consistently noted difficulties with opposing views, styles of writing, schedules, or expectations based on personality. Interactions of this nature primarily centered on the deliverables, including the best way to summarize or present our research. In reflecting on these instances, one student stated, "I found that we had quite a few strong personalities in our group and that this was a potential issue. Everyone was so passionate about getting their ideas out that sometimes these ideas clashed". This was especially evident throughout our brainstorming sessions, where certain group members were more vocal about their ideas regarding the visual format of our work come to fruition.

### 7.4. Diversity of Community University Partners

Further to the collaboration among students, the diversity of the CUP membership posed significant benefits and challenges to working as a team. As noted earlier, our team comprised students from various disciplines (including sociology, geography, psychology, criminal justice and public policy, and political science), and included both Ph.D. and Masters level students.

Throughout our reflections, team members consistently acknowledged the unique skills, perspectives, and area of expertise brought to the table that might otherwise be overlooked in single-disciplinary papers or partnerships (Ramaley 2014). For example, one student identified the team's diversity as "the most beneficial aspect of this collaboration. Specifically, our diverse backgrounds allowed us to contribute unique perspectives, and address certain limitations or shortcomings that we might have otherwise overlooked". Another student described the diversity of the team as "a rousing success", whereby "an interdisciplinary team has given us the space to think on our feet and pivot to alternative deliverables, processes, methods, goals, and products".

As mentioned above, working as a team posed challenges, many of which were arguably exacerbated by our diverse backgrounds. As one student noted, " . . . because academic disciplines are all unique and have their own ways of doing things, this was also a challenge that we had to overcome. For example, there are differences in writing styles and knowledge mobilization tool preferences between fields like sociology, psychology, and geography". To further elaborate, one student explained:

> At times, the writing was somewhat disjointed in terms of the content being included and the style being used to get our message across. It was clear that students from one discipline valued more tangible/quantifiable evidence (including numbers, graphs, charts, etc.), whereas students from other disciplines valued more qualitative evidence gathered from existing literature. Thus, in some instances, there were issues in deciding which academic and presentation style would be the perfect fit for communicating our experiences.

Others noted the difficulty of incorporating all or equal contributions from students at the Ph.D. and MA levels. For example, the MA students did not all share the same research experience as the Ph.D. students, including searching for, reading, reviewing, and synthesizing academic literature. Another example of the difficulties in having both Ph.D. and MA level students was the varying levels of experience in creating knowledge mobilization tools outside the traditional format. As a result, there were feelings of inexperience that the students ultimately had to overcome, and instances where they felt they had to compensate in creative ways. For example, one of the MA level students reflected on having minimal experience in creating outputs that would be useful in non-academic settings. In reflecting on this perceived limitation, the student stated, "in comparison to some of the Ph.D. students, there were times where I felt that I had much less to contribute to the development of our knowledge mobilization tool, especially because I had never created anything in this format. Because of this, I felt that it was important for me to do some information gathering on the various types of knowledge mobilization pieces and about other ways of disseminating information so that I could contribute to the group".

However, these varying levels of experience were also identified as beneficial, as they led to a coming together of the group in order to work through each student's inexperience. For example, one of the Ph.D. level students noted how "this juxtaposition was ultimately complimentary, as it allowed us to build on, and address weaknesses or limitations in, each other's perspectives". Specifically, it was through these opposing views that we successfully learned how to enact and embody several of the principles of CES, including: collaboration, balancing of power, reciprocity, and clear communication.

### 7.5. Creative Thinking

As previously mentioned, many of our team members typically produce deliverables in the form of written articles. Therefore, this class introduced us to the concept of knowledge mobilization[4]. Accordingly, we were faced with the challenge of thinking outside the box and designing deliverables that would assist our community partners in educating their target audiences, and in effect, reach large and diverse audiences. Collectively, our team found this to be a particularly rewarding experience, whereby one student stated, "One of the most influential aspects of this course was learning how to communicate information to audiences in a way that is interesting and accessible". Further, many members of our team suggested that the knowledge and experience obtained in creating our knowledge mobilization pieces would be further utilized in future research endeavors (Abraham and Purkayastha 2012). These types of CES learning opportunities are of value across disciplines. In having a class with students from multiple backgrounds, thinking outside our disciplinary boxes and working in creative ways provided valuable experience. As one student stated, "I now recognize the value and importance of thinking outside the box, and presenting information in a way that is interesting and accessible to multiple audiences. As such, I plan to continue practicing my creative thinking, and incorporating my new understanding of knowledge mobilization throughout my future research endeavors".

### 7.6. Meaningful and Applicable Research

A final benefit identified by each team member is producing meaningful and applicable research (Gelmon et al. 2013a). The overarching purpose of our research was to produce deliverables that our community partner could utilize as part of their education and advocacy to address the prevalence and seriousness of violence against women in Ontario.

One student expressed how pleased she was that we (the students) were able to use the research to produce a KMb product (see Figure 1) that would be meaningful and useful for the community

---

[4] The purpose of knowledge mobilization is to ensure that all citizens benefit from publicly funded research. It can take many forms, but the essential objective is to allow research knowledge to flow both within the academic world and between academic researchers and the wider community. By moving research knowledge into society, knowledge mobilization increases its intellectual, economic, social and cultural impact (SSHRC 2014) (Cooper et al. 2018, p. 2).

partner: " . . . we were able to come up with an idea that is visually appealing and accessible, so that the general public (as well as other service providers) can become educated on misrepresentations occurring in mainstream news".

**Select Media Category from drop-down**
All

| Femicide Victim | Media Category | Media Source | Victim Humanized | Picture of Victim | Gendered Social Problem | Labelled a Femicide | Victim Blaming | Individualized | Voice of Authority | VAW History Undocumented |
|---|---|---|---|---|---|---|---|---|---|---|
| Ardis, Margaret | Local Paper | Windsor Star | G | | | | | | | R |
| | National Paper | National Post | G | | | | | | | R |
| | TV Media | CTV News | | | | | | | | R |
| Baechler, Lorrie Lynn | Local Paper | Kitchener Record | | | G | | | | | R |
| | National Paper | None | | | | | | | | |
| | TV Media | CBC News | G | G | | | | | | R |
| Baran, Candace | Local Paper | Sudbury Star | G | G | | | | R | | R |
| | National Paper | Toronto Star | | | | | | | | R |
| | TV Media | CBC News | | | | | | R | | R |
| Bennett, Kristina | Local Paper | St. Catherine Stand… | G | G | | | | | R | R |
| | National Paper | Toronto Star | G | G | | | | | | R |
| | TV Media | Global News | | | | | | | | R |
| Brar, Gurpreet | Local Paper | Brampton Guardian | | | | | | | | R |
| | National Paper | National Post | | | | | | | | R |
| | TV Media | Global News | | | | | | | | R |
| Campbell, Eleanor | Local Paper | Toronto Sun | G | G | | | | R | | R |
| | National Paper | National Post | | | | | | R | R | R |
| | TV Media | CTV | | G | | | | | | |
| Charbonneau, Precious | Local Paper | Toronto Star | G | G | | | | R | R | R |
| | National Paper | National Post | | G | | | | R | R | R |
| | TV Media | CBC News | | G | | | | R | R | R |
| Consuelo, Sylvia | Local Paper | Toronto Sun | | G | | | | R | R | R |
| | National Paper | Toronto Star | | | | | R | R | R | R |
| | TV Media | CBC News | | G | | | R | R | R | R |
| Cooper, Melissa | Local Paper | Toronto Sun | | G | | | | | R | R |
| | National Paper | Toronto Star | G | G | | | | | R | R |
| | TV Media | CBC News | | | | | | | R | R |
| Guimond, Nicole | Local Paper | Ottawa Citizen | G | G | | | | R | | R |
| | National Paper | None | | | | | | | | |
| | TV Media | CBC News | G | G | | | | | | R |
| Kochie, Shannon | Local Paper | North Bay Nipissing | G | G | G | G | | | R | R |
| | National Paper | None | | | | | | | | |

**Figure 1.** A sample screenshot from the knowledge mobilization (KMb) tool produced for the community partner. The "heat map" outlines positive and negative frames identified in different media reports for each of the 29 femicide victims in the years 2015–2016. A green or red diamond represents evidence of positive or negative framing respectively in that particular news report. On the full interactive version, available on the community partner's website [See: https://www.oaith.ca/oaith-work/current-initiatives.html], users can select between local, national and TV media sources from the drop down menu found in the top left corner of the map. Additionally, clicking on the red or green diamond takes the user to the original media reports analyzed for this research product.

In contrast to academic papers, many team members enjoyed contributing deliverables with real world applications and tangible outcomes. One example of the value of this real world application is identified in the following student's reflection.

The most rewarding part of this course was getting to participate in [the provincial not for profit association] Provincial Training Day. It allowed me to have conversations with

professionals working in the field, which led to lots of educational moments. In addition, it gave me the opportunity to practice my presentation skills in a practical (rather than academic) setting.

The overarching positive experience of the students throughout this process and the enjoyable experience of creating deliverables outside the traditional academic paper format speaks to the otherwise mundane and routine experiences of traditional academia. Specifically, it becomes quite evident that being unable to see real-world applications of our work has negative implications on students. For example, while producing a research paper may strengthen students' writing proficiencies, it arguably overlooks the importance of learning how to communicate in different ways and to engage audiences outside of academia. In our case we had the opportunity to work on a broader range of skills that can be of equal importance, such as: learning how to facilitate working with a community organization outside the academic institution, creating and implementing a 'Plan of Action' for meeting a community need, and creating visually appealing user-friendly research dissemination tools. Within our student reflections, we continuously outlined the fact that there are not many opportunities for students to participate in courses where they are able to create deliverables that have real-world implications. Thus, many students are not adequately prepared with the skill set that will allow them to excel in venues outside the university. Further, when students are unable to see how the work they did actually contributes to others in the 'real world', it makes the work less meaningful. Therefore, one of the greatest take away points from this process was that students thoroughly enjoyed taking part in the type of course that provides real-world benefits to the community. In reflecting on these outcomes, we propose that these types of courses and this mentorship would be a welcomed addition to any university institution.

## 8. Community Partner Reflections on Our Community University Partnership

### 8.1. Evolution of the Project

One of the benefits of this process was the evolution of the project over time. When the faculty member and community partner began the collaboration four years ago, the aim of the project was mainly to create a femicide list to memorialize women who had been murdered by men each year in Ontario. However over time, the project significantly progressed, and the outputs of the project began to change. Specifically, the project shifted focus to synthesizing research on media framing of violence against women and femicide, and examining mainstream media reports of the way the femicide victims in a given year had been depicted. The purpose was to determine whether such depictions corresponded with the research on media framing, and to compare the information that came from mainstream media about femicide cases with more formal sources, including reports from the coroner's office. The added components of the research project also resulted in new and unique knowledge mobilization outputs, and the level of engagement with the students also began to increase. For example, we were able to work on an analysis on the femicide data and not only identify and document femicide victims. This required more consultation and collaboration between the students, the community partner, and the faculty member.

Ultimately, the community partner noted how "the progression has been pretty substantial over the past 4 years". As a result of the additional work on femicide conducted by the CUP, there have been greater opportunities for education and advocacy related to femicide and violence against women for the partner and their networks throughout the *entire year* (not just September and November—the key times where the annual list of women who have been murdered are represented on 6 December). For example, the community partner noted, " . . . this year we used other knowledge mobilization tools that we were able to use throughout the year and were able to release those at different times throughout the year". This included releasing some of the significant research findings via Twitter.

### 8.2. Diversity of Students Involved

Similar to one of the benefits identified by the students involved in this project, the community partner also argued that the students' diverse disciplinary backgrounds had a positive impact on the research. She stated the following:

> [Students] were coming from different disciplines. I think that has been a huge benefit. They are really able to look at the information from different angles. With this project, having folks from a geography discipline has been a huge advantage. We were looking at mapping of femicides and experiences, so having people from that background was certainly beneficial in ways [ . . . ] we hadn't had that discipline before. We had psychology and sociology in there too. They understand some of the impacts—social, emotional impacts and they have a bit of a contextual understanding.

The community partner found that having students from a variety of disciplines led to diverse knowledge and experience that benefited the research and the collaboration. If the discipline of students involved was the same each year, there would not be "space for things to evolve and change. If we keep it in the standard format every year we would probably come up against a lot of roadblocks. Students come from different disciplines and different knowledge and experience—all of that really does have an impact and a role on what we're asking them to do".

### 8.3. Mutual Benefit

The community partner recognized the mutual benefit for all members involved in the partnership, whereby, "Everyone had an opportunity to contribute their area of expertise and felt valued in that process. People really focused on what their strengths were and engaged in aspects of the partnership and the project based on their strengths. Some were great with coming up with graphics, some great at putting data sets into tables and maps".

An important component of mutual benefit in CES is related to outcomes and impact. Increasingly, CES is evaluated in relation to what the CU partnership was able to create together that resulted in mutual outcomes and meaningful impact. The outcomes of the project were beneficial to each of the members in the CUP, not only the students, faculty member and community partner, but also the wider society. Specifically, the community partner identified that because of the students' awareness of various ways of conveying the project's messages for different venues (for example "online spaces that were used on social media to translate some work they were doing, conferences, other meetings"), the information was disseminated in a variety of ways, and for diverse audiences, which was one of the intended outcomes for the CUP.

Furthermore, the community partner identified that there is a social change outcome that is a benefit to the wider society. She stated this in the following way:

> Because the project is ongoing, it doesn't really have an end. It is unique in that we have to constantly evaluate how things are going, measure effectiveness because women continue to be murdered. Every year that continues, and the work to identify those issues will continue. We have to look at this in a long-term way and understand that social change isn't going to be achieved in one semester. Rather, it is contributing to what has already been done and helps to inform next steps.

### 8.4. Communication and Miscommunication

In addition to highlighting some of the benefits of the CUP, the community partner was able to reflect on the overall process and identified that there were some obstacles and challenges along the way. For example, miscommunication between the partners impacted the progression of the project at times because we did not have the kind of real time collaboration tools in place for everyone in the partnership to see what was being worked on or what progress had been made. Moreover, she

explained that early in the CUP, we did not have as rigorous a methodology as we do now for the kind of collaborative data collection, coding, and analysis and this limited our ability to document and communicate to others what we did and why we did it. We also did not have the kind of communication strategy and tools to use between students, as well as between students, the faculty member and the community partner. However, throughout this term, there were greater connections and communications between the community partner and the students. This included " ... meetings over the phone, via Skype, in class with me coming into the classroom and students coming to a conference organized by our organization [ ... ] these opportunities are important and they played a positive role". The graduate students also created an automated article search engine (Mi Data Labs(MDL) 2017) that develops a database of articles relevant to femicide by automatically searching through Ontario and national news websites every day. The database can be examined for further analysis by student researchers at a future date.

*8.5. Partnership Development*

The community partner identified that the challenges or obstacles that arose (also including some organizational issues) were most prevalent throughout the first few years of our partnership. She indicated that, "There has been so much growth and innovation and really figuring out a way to understand this information, there's been a lot of positive. It has taken 5 years to get to this point [ ... ] but this really shows that this type of partnership evolves and it takes time and you have to give that time".

As a result of these obstacles and challenges, the community partner identified some changes she would make if given the chance to undertake this project again. Specifically, she would like to be able to include a long-term planning process and schedule. This would involve determining such dates as when she would be present in the class sessions. Further, she would aim to "look at the project in a long-term way versus on the short-term. It is a long-term partnership. As long as it's working, we hope that it can continue. It is something that's constantly evolving".

## 9. Faculty Reflections as a Community Engaged Scholar

The above reflections mirror and reinforce the academic literature on CES and other experiential pedagogies as well as my experience with previous CUPs doing CES and community engaged learning (CEL) projects. Below, I identify and reflect on some of the benefits and challenges that speak to the principles and practices that we deliberately and intentionally attempted to model.

*9.1. Scarcity of Time: Scholarship, Relationships, and Planning*

CCES is time consuming (Holland 1999). As a CE scholar working in a traditional academic institution and department, I find it extremely difficult to generate outcomes and products that balance community priorities and traditional academic requirements (Murphy and McGrath 2018) for co-knowledge generation, transmission, and application. "The pressure on faculty members to produce peer-reviewed journal articles and continually chase grant funding, what Smith (2010) refers to as *academic treadmills,* is well documented (McGrail et al. 2006; Smith 2010)" (Cooper et al. 2018, p. 4).

In this specific case, trying to conduct CES with students within an academic calendar (i.e., 12 weeks) is a challenge (Strier 2014; Klein et al. 2011). While I regard CES principles as a necessary foundation for ethical, sustainable, and successful CCES, they are nevertheless difficult to fulfill, especially within an inflexible institutional timeframe. For example, developing genuine and reciprocal relationships, deciding on partnership roles and responsibilities, and establishing and using collaborative processes and tools to work together toward shared goals takes time (Morton 2013). Accordingly, one of the most significant challenges includes finding the balance necessary to complete this work in a principled way and scoped appropriately to deliver on the outcomes the CUP has agreed to, while remaining flexible to changing circumstances (Metzger 2012).

It is common for me to engage with a community partner where there is an established relationship. It is helpful for a number of reasons (Warren et al. 2018), but the challenge still remains for the students and the community partner to develop a relationship with one another. This has implications for embracing other CES principles, including: communication, collaboration, balancing power, and meaningful outcomes. For example, as mentioned above by both the community partner and the students, communication, and collaboration was challenging. I continue to think about how to provide more opportunities, in addition to what I normally do, for the students and the community partner to have direct contact and get to know each other, without overburdening the community partner, and without taking too much time away from other course requirements and project activities. I look for ways to work through and negotiate roles and responsibilities, identify each other's strengths and capacity, and work toward agreed upon goals while staying open to the possibility and likelihood that "plans" will change once we get into the work (Northmore and Hart 2011).

### 9.2. Flexibility, Managing Change, and Knowledge Mobilization

CES best practices speak of the iterative nature of co-creating knowledge (Nichols et al. 2013; Nichols et al. 2014; Watson-Thompson 2015; Janzen et al. 2017). This requires flexibility, the ability to make changes as you go, without losing rigor and scholarly processes and to expect the unexpected. Each time I embark on a CES project with a CU partner that includes students, I am reminded of how different I experience change and the unknown, compared to my students. This is what I often refer to as the "muddy, messy, and missing" aspects of CES/CEL (Morton 2013, p. 7). The graduate students referred to this as "uncertainty" in their reflection. Because I now expect this to be experienced as a challenge by students, I have developed a specific course learning outcome that requires students to demonstrate their ability to "Manage individual and collaborative teaching and learning experiences in changing circumstances". I added this learning outcome as one way to intentionally and explicitly alert students to the fact that they will experience this uncertainty and will likely find it stressful. I find this challenging to teach because I have not found processes and/or teaching and learning activities that help build this skill appropriately. This tension is normally experienced at two points in time: at the front end of the project when students are feeling unsure about how the project will take shape and what will be required, and at the back end of the project with respect to knowledge mobilization. Part of the tension is that students are often frustrated that I do not lay out, with greater clarity and specificity exactly what we will be doing, and what it will look like in the end. I have to remind them that although we have project goals, these goals can be accomplished in myriad ways, and our job is to use our collective knowledge, skills, and resources to map it out, and carry it out, but also reflect and adjust as we go. This is usually foreign to students given their previous educational experiences. They are not used to being asked to do as much collaboration and problem solving and they are not used to communicating research via non-traditional academic outputs. In our case, it was not appropriate to write a research paper for the audience we were targeting, and yet, that is the type of scholarship/output that students were most comfortable producing. I find this both exciting and discouraging. On the one hand, CES provides students with the opportunity to develop and hone academic and community facing knowledge, skills, and values to address and solve complex social justice issues with their communities (Levkoe et al. 2014). Yet, I know that neither I nor the undergraduate or graduate program offers enough time, resources or support to fully help students develop the unique knowledge, skills, and values that CES requires and on which they are being evaluated. This is a problem that will be returned to in a subsequent section.

### 9.3. Trust, Collaboration, and Power Dynamics

Aligned with the principle of "mutually beneficial relationships", CES requires trust from all partners (Davis et al. 2017). It requires trust in a process that is usually foreign, and/or, sometimes based on previous negative experiences from both the community partner (Blouin and Perry 2009; Levkoe 2017) and the student perspective. Beyond the pressures of time, students report that they

have either no experience or negative experiences with "group work" (Pauli et al. 2008). Even for those that do have some experience, the collaboration is unlikely to include their professor and a community partner and it may or may not have been formally assessed. The institutional requirement of grading is a significant power issue that is particularly challenging within a CEL environment. CEL takes time, trust, support, practice and resources to work collaboratively and authentically. This challenge embraces issues of power (Davis et al. 2017; Sandmann and Kliewer 2012; Way 2013) between students and between students and the faculty member and community partner. The reality of conducting CES within a course, or as part of a student's program, means that the faculty member and often student peers are evaluating student knowledge, skills and values. To help lessen undue pressures that may result from such an evaluative environment, it is important to offer formative assessments in the form of continuous constructive feedback and not just summative assessment (i.e., ranking student performance at the end of a learning cycle). This is especially important when we are expecting students to demonstrate unique learning outcomes or learning outcomes associated with unique activities or assignments previously foreign to them.

In our course, for example, students were assessed on their teamwork and collaboration skills. However, most students were unfamiliar with using collaborative tools like work plans or Google Documents for the purpose of collaborative writing and editing. This unfamiliar territory created tensions, particularly when students' efforts and contributions were assessed within a traditional academic context. Close to 40% of the grades in this course were connected to the students' ability to work and write collaboratively toward community partner outputs including unique knowledge mobilization. There is a lot to be said about the challenges of student assessment in CEL and while this is beyond the scope of this paper, it is an area about which my colleagues and I have done a lot of thinking. For example, opportunities for formative assessment (Fluckiger et al. 2010) at the beginning and not just at the end of a learning cycle (i.e., summative) are important, and scaffolded assignments help to reduce student stress because each piece of work is worth less and therefore is less risky given the grade impact. Throughout this course, formative assessments included oral presentations, critical reflection assignments and group work (Oakley et al. 2004). Summative assessments included a community partner report and knowledge mobilization tools. See (Koh et al. 2012; Ashford-Rowe et al. 2014; Fluckiger et al. 2010; Hermon et al. 2013) for other relevant literature on assessment. Half of this course (50% of the grades) required students to apply the content they were learning and skills they were developing or demonstrating to "meaningful and relevant tasks" (. Litchfield and Dempsey 2015, p. 69) associated with the femicide project and therefore would be referred to as Authentic Assessment (Ashford-Rowe et al. 2014; Azim and Kahn 2012).

Other power issues were also present between the CUP and this proved to be challenging to address. For example, as mentioned above, some students struggled with "handing over" their work and not knowing how the community partner might use it in the future. I recognized that this was an area I had not created enough space to talk with the CUP about, nor was it explicitly addressed at the beginning of the partnership. From my perspective, and in keeping with the principle of addressing a community identified need, I felt strongly that beyond acknowledging the CUP's work, that we were working to develop a knowledge mobilization tool that was for the community partner to use as needed and that it would have a life of its own once created. However, it was an important conversation to have had much earlier in the term. The tension points had to do in part with students wanting to maintain their intellectual property and their concern about their work being used in ways that they might not agree with in the future.

Despite the challenges that existed for me in this CU partnership specifically, and in CCES in general, I am a strong advocate of CCES and argue that the potential benefits for all members of the CU partnership and the larger community can outweigh the challenges. To go further, I argue that this kind of course and work in higher education is imperative. Korzun et al. argues that "[t]he CES model encourages an exchange of resources between all members of a CU partnership in an effort to identify and address solutions to community and societal problems (Davidson et al. 2010; Edelglass

2009; Showalter 2013; Strand 2000)" (Korzun et al. 2014, p. 110). Effective CES also demands that scholars produce diverse forms of scholarship in innovative formats (Gelmon et al. 2013a). If done in a principled way, and using best pedagogical practices, CES can help prepare students and faculty to work with their communities to address and potentially solve complex social justice issues (Tarantino 2017; Warren et al. 2018; Ramaley 2014; Silka et al. 2013).

## 10. Lessons Learned

Despite our differing experiences and unique ways of articulating benefits and challenges, we all agree that the benefits of this project, including the lessons learned and what we produced together (i.e., a knowledge mobilization tool to be used annually as a way to analyze the way Ontario femicide victims are portrayed in mainstream news) outweighed the challenges.

As a community partner, we now have a well-developed process, an evidence-based coding instrument and media framing tool, and meaningful outputs including knowledge mobilization products (i.e., the interactive Heat Map and aggregate analysis) that are both on our website[5] and is an important part of our ongoing education and advocacy about femicide and violence against women. This is significant because we know that media portrayals of femicide and femicide victims impact the way the issue of violence against women is understood and acted upon. Therefore, we wanted to create a research-based, engaging and impactful analysis accessible to the general public, violence against women advocates, politicians, and journalists.

As graduate students, this experience changed the way we worked with each other, our community partners, and the instructor. We moved beyond peers in a graduate sociology class and instead became emerging community engaged scholars working together on a femicide project and hoping to create something that would challenge the way femicide victims are portrayed in mainstream news. We learned how to communicate and collaborate in new and unfamiliar ways and experienced the challenges and benefits of co-creating knowledge across different disciplines, skills, and personalities. We learned how to communicate and disseminate co-created knowledge in ways that required a combination of knowledge and skills that we had previously possessed and had to develop. Our traditional definition of "scholarship" was challenged, as was our assumptions about the extent to which CCES is rigorous (Warren et al. 2018). We have a greater appreciation of the complexity of the processes required to carry out CES and the importance of CCES principles as a foundation for the work.

As a faculty member, this CU partnership impacted my approach to teaching, learning, and training graduate students in CCES. Based on this experience and our reflections of this experience, I am more convinced than ever that higher education has a responsibility to students, to our communities, and to the planet to train students in active and experiential pedagogies in order to work on real world social problems. In particular, higher education has a responsibility to train students in CES and CEL so that they are better positioned to work together as students, community partners, and faculty to address and attempt to make positive social changes to complex social issues. I have made significant changes to this graduate course (principles and practices of CCES) based on what I learned. For example, I now actively invite and encourage students outside of sociology to register for the course. Although there are important considerations to be aware of and address with diversity in the classroom (Lyon and Guppy 2016), the benefit of working with students from multiple disciplines outweighed the challenges it created (Silka et al. 2013). I have also adjusted the learning outcomes, the teaching and learning activities throughout the term, and the nature of the assessment to better reflect the collaboration as well as knowledge mobilization skills that are required (Nichols et al. 2013). My increased awareness of the challenges of collaboration and knowledge mobilization, especially in a

---

[5]　Please see the following website to see the live and interactive KMb tool on the community partner's website: https://public.tableau.com/profile/midatalabs#!/vizhome/OntarioFemicideMediaAnalysis2017/Story1.

traditional academic context, have led me to provide more resources, support and be more explicit and intentional about these challenges. Finally, the work we did in this project has also impacted my undergraduate teaching and learning. Specifically, for the last three years, I have used the femicide media framing and analysis process, heat map knowledge mobilization tool, and aggregate analysis visual developed in this CU partnership as the basis for a community engaged learning project in a 4th year undergraduate sociology class. This means that the partnership between the University of Guelph and OAITH continues to provide mutual benefit and meaningful outcomes for all members.

## 11. Insights and Offerings for Others Interested in Pursuing Critical Community Engaged Scholarship

In reflecting upon both the challenges and benefits associated with undertaking CCES, there are several insights and offerings we have for future projects, and for students, faculty members, and community partners who are emerging community engaged scholars, or considering this approach for the first time. These insights are also relevant for higher education administrators. While we all agree that conducting CEnR in this partnership was worth the struggles we experienced, we highlight three main recommendations for others considering participating in and/or designing and supporting CEnR, especially within an academic course. These include: (1) a commitment to foundational community engaged principles as a guide for practice; (2) advocating for institutional support in the form of material, personal, and other awards and rewards; and finally (3) we recommend making visible and explicitly demonstrating ways in which CES and CEL satisfy the criteria required for quality scholarship. Any kind of CES should be guided by common foundational principles outlined above. Furthermore, CES should be conducted in a way that critically reflects on how its processes and outcomes support equity and aim to dismantle systemic sources of injustice at the classroom, institutional, and community level. Recent research has identified reciprocity, community-identified needs, and high-quality scholarly investigation of real-life social problems as the most common and important principles for effective community–academic research partnerships (Beaulieu et al. 2018; Gordon da Cruz 2018; Dave et al. 2018). There is now an abundance of research, tools and resources available to support faculty and higher education institutions interested in designing and offering courses and other pedagogical activities which engage students, faculty, and community partners. (See resources such as Community Campus Partnership for Health, https://www.ccphealth.org/; Community Based Research Canada's Community Based Research Excellence Tool (CBRET) https://communityresearchcanada.ca/cbret/, University of Victoria https://www.uvic.ca/cue/) One simple piece of practical advice for working toward these principles includes co-creating a work plan using a Google Document so that all members have access to draft, edit, comment, and collaborate toward the development of a set of goals, activities, and timelines in real time.

Our next recommendation is related to the issue of risk and reward. Given the amount of risk and uncertainty that students, faculty, and community partners often experience in engaging in community university partnerships, especially when embedded in courses, institutional structures supporting and rewarding such engagement are important for these efforts to be successful and sustainable. Institutional support must include leadership and support from deans, provosts, and presidents, both for institutional change and for individual faculty (Cavallaro 2016). Further, institutional support means framing faculty work within the context of the institution's mission which often includes a commitment to "serving society" and assessing the actual work in which faculty are engaged and committed (Calleson et al. 2005, p. 318; Franz 2009). This requires changes to tenure and promotion policies that redefine definitions of "scholarship" and recognize and reward CES scholars for a spectrum of scholarship beyond traditional measures of impact such as academic journal articles and books (Calleson et al. 2005; Gelmon et al. 2013a). It also means valuing and rewarding time spent in community partnership formation which is important for building relationships and trust. When possible, we encourage future CES scholars to seek out and use the institutional infrastructure and support that is available within higher education institutions to support principled engagement. For

example, this can include seeking out research services assistance in applying for community engaged research and training funding. This can include working with administrators to access classroom space that is more conducive to the pedagogy (e.g., adequate and accessible meeting space for CU partnerships and for weekly student collaborative group work). It can mean taking the time to work with librarians, and knowledge mobilization experts to provide specialized knowledge and skills training that neither students nor faculty have, in order to produce scholarship that is more engaging and likely to have greater impact. It can include working with teaching and learning experts in institutional teaching and learning centers and in particular experts in community engaged learning for support via resources such as rubrics, assessment examples, teaching and learning activities as well as being invited to participate in communities of practice with others who are doing CEL. And if such support is not available, we need to ask for it and encourage institutions to learn from those who are leaders in supporting and rewarding CES.

A third offering is to be prepared for critiques that suggest that using CES to support social justice agendas and to work with communities to advocate for change is less rigorous than more "traditional" scholarship (Warren et al. 2018). Although this was not directly addressed in the above reflections, it was something that the students and the faculty member encountered and continue to face. During our class discussions, several students shared examples of past experiences with faculty undermining CES and other "applied" research endeavors. We found it helpful to encourage each other to stay motivated to continue conducting and improving the rigor of our CES work by discussing responses to such criticisms from the CES literature. Specifically, we think Warren et al. (2018) provided an important response to this critique and we refer to Jordan et al. (2009) who identify the eight characteristics that are used to evaluate quality CES. We agree with Jordan et al. (2009) and Warren et al. (2018) that rigor can be strengthened through substantive community collaboration. In fact, following processes and practices within a CES framework provides clear guidelines for demonstrating and documenting quality and rigor while simultaneously co-creating and mobilizing knowledge that helps to advance social justice.

## 12. Conclusions

Our critical community engaged scholarship and the reflections on community engaged learning with our community university partnership contribute to the research-teaching-service nexus via CCES. Since Boyer's (Boyer 1990, 1996) call for a broader view of scholarship as an integration of research, teaching and service, CES has the potential to unite and integrate these three core missions (Stanton 2000). Gelmon, Jordan, and Seifer suggest that applying institutional resources to solve problems facing communities through direct collaboration with those communities using accepted standards of scholarship " . . . educates students for democratic citizenship, mobilizes multiple forms of knowledge, and leverages the capacities of all the participants to improve community well-being" (Gelmon et al. 2013a, p. 59). Still, despite the potential benefits that exist for students, community partners, the larger community, faculty, and higher education institutions (Olberding and Hacker 2016; Bruning et al. 2006), this work is not without its challenges, limitations and critiques. In order to be aware of and intentionally address both the obstacles and the opportunities within the research-teaching-service continuum, the well-known and documented principles and practices aligned with CCES must now be embraced in a reflexive way. Doing so offers the best hope for successful, ethical, and sustainable CU partnerships to use a CCES framework to produce meaningful outcomes for the purpose of social justice. In our CCES example, we addressed the local and global social justice issues of violence against women and gender equality more generally. Such critical issues require the kind of community engaged research and action our collaboration reflected on from our positions as students, community partners, and faculty.

**Author Contributions:** Conceptualization, M.M.; Data curation, M.M., A.S., C.S., and A.W., E.P.; Formal analysis, M.M. and A.S.; Methodology, M.M.; Project administration, M.M.; Writing—original draft, M.M., A.S., C.S., A.W., and M.H.; Writing—review and editing, M.M., A.S., E.P., A.W. and C.S.

**Funding:** This research received no external funding.

**Acknowledgments:** Special thanks to Abhilash Kantamneni and Alexa Mackenzie-Cooper who were students in the course but unable to participate in writing this paper.

**Conflicts of Interest:** The authors declare no conflict of interest.

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
