# Peer review of "Graduate Students, Community Partner, and Faculty Reflect on Critical Community Engaged Scholarship and Gender Based Violence"

_socsci, doi:10.3390/socsci8020071_

Round 1
Reviewer 1 Report
First, I want to congratulate the authors of this paper for undertaking this original research project, but also for putting significant effort into translating this work into a peer-reviewed manuscript. The community engaged research and learning that has been accomplished is innovative and important, and clearly demonstrates what can be accomplished when the principle of applying institutional resources to real world problems is put into action. The suggestions I have below are considerations for the authors in order to build even further on the existing strengths of the article.
Embrace the innovative nature of the work even further by emphasizing the significance and value of undertaking community engaged research and learning. The article certainly does a thorough job of locating the work within the appropriate literature, but might there be an opportunity to push an argument further that this type of course is actually imperative to challenge traditional notions of research and learning inside of post-secondary institution? It struck me as I was reading student reflections about how stretched they felt - and at times even overwhelmed - that there might be an argument to be made that as educational institutions we are collectively failing students because we continue to develop a narrow set of skills? And that we need to think differently at all levels of academia about how to develop diverse skills and knowledges as part of our educational mission? Similarly, are we failing the communities that we are a part of because knowledge so often continues to benefit those inside of academia rather than the broader public? The authors do not need to argue these things specifically, but rather I wanted to provide them as examples to consider the significance and value of this case study, and making a clearer case for why others in academia should sit up and take notice!
Increased specificity in the reflections and descriptions of how the project was experienced. There were a number of times when the reflections would benefit from more information and details. For example, line 406 states that "working as a team allowed us to build on each other's ideas" and line 445 states "others noted the difficulty of incorporating contributions of individuals at the PhD and MA levels". While I appreciate that authors may be navigating how to talk about potentially sensitive scenarios, statements like these leave the reader with a vague impression that feels relatively surface level. What does it mean to say that you "built on each other's ideas?" how did this happen? what does "building" look like? perhaps there is an example that could illustrate this further. Similarly, what were the specific challenges in related to incorporating contributions from individuals at different levels of graduate study? How did these difficulties manifest? Is there anything more specific that can be said to give the reader a more clear picture? Overall, this was most apparent in the student reflections and providing more specificity throughout would benefit the reader. There may also be opportunities to increase specificity in other sections as well. For example, line 565 notes that the community partner stated that "miscommunication impacted progress" --- in what ways did it impact progress? how much? what did this look like? Of course, this needs to be balanced with the length of the manuscript, so it will likely need to be strategic.
Further development of the insights/recommendations. The insights/recommendation section seems to be a central aspect of the paper, especially for those who are reading the work with the hopes of putting some of the "lessons learned" from this experience into action in their own classrooms and institutions. Yet, as a reader I was left wanting much more. Is there a way to be more concrete in this section and develop the ideas further. For example, line 751 suggests to seek out institutional infrastructure and support, but what might that even look like? What type of infrastructure and support is needed? Additionally, are there any further insights to provide around the size of the research project (how big to make it? how to put boundaries around it?) or the type of community group to work with? and so forth. It is possible that a lot of these ideas are already embedded throughout the paper, but could be drawn out more explicitly at the end. For example, the faculty member noted that they already had an existing relationship with the community group - perhaps a recommendation is to only work with a group that you already have an existing relationship with? The faculty members also mentions feeling frustrated and unsure about how to develop certain skills in the classroom context - perhaps an insight might be to reach out to institutional teaching centres to help create activities to develop these skills?
Address the institutional requirement of grading. The need to provide students with a grade at the end of the course seems to be an issue that was underlying a number of the tensions and challenges, but grading was never explicitly discussed or addressed. How did the instructor grade the students? I can imagine there are many challenges given the constraints of the existing system. And how did this impact the student's discomfort with this type of a course? Moreover, what does this mean for the power dynamic between the students and the professor, the students and the community partner, and potentially the professor and the community partner? If one of the key aspects of community engaged research and learning is to think about power, it seems like the institutional requirements to grade students in a particular way creates a number of challenges (and opportunities??) for the principle of balancing power.
Author Response
Dear Reviewer 1. Thank you for your thoughtful feedback. Please see the following responses to your suggestions/recommendations.
Embrace the innovative nature of the work even further by emphasizing the significance and value of undertaking community engaged research and learning.
To address the suggestion to make a clearer case for why others in academia should sit up and take notice we added more about the overarching positive experience and value of this project for each of the students. Specifically, we added the following:
“The overarching positive experience of the students throughout this process and the generally enjoyable experience of creating deliverables outside the traditional academic paper format speaks to the somewhat mundane and routine experiences of traditional university academia. Specifically, it becomes quite evident that being unable to see real-world applications of our work has negative implications on students. For example, while assigning students the task students of conducting a research paper and presenting that research in a traditional essay/paper format may strengthen their, traditionally academic proficiencies, it arguably overlooks a broad range of skills that can be of equal importance, such as learning how to facilitate working with a community organization outside the academic institution, creating and implementing a ‘Plan of Action’ for meeting a community need, and creating visually appealing-while at the same time user-friendly tools. Within our student reflections, we continuously outlined the fact that there are not very many opportunities for students to participate in these types of courses, where they are able to create deliverables that have real-world implications and so, students are not adequately prepared with the skill set that will allow them to excel in venues outside the university. Secondly, when students are unable to see how the work they did actually contributes to others in the ‘real world’, it makes the work less meaningful. Therefore, one of the greatest take away points from this process was that students thoroughly enjoyed taking part of courses such as this because not only do they have personal rewards for the students but they also provide real-world benefits to the community. As a result, these types of courses, should become commonplace within the university institution.
This section was added in order to address the reviewer’s comment about the potential of educational institutions failing students because students continue to develop only a narrow set of skills. To address the comment about the fact that we are failing our communities because often the work done at the university benefits only those inside academia and not the broader public, we added more specific examples to the Lessons Learned and Conclusion about how important this work is for addressing complex social problems.
Increased specificity in the reflections and descriptions of how the project was experienced.
To address the above request, we added more concrete examples within the student reflections. Specifically, we added an example of how we ‘worked as a team’ through building upon each other’s ideas, especially with regards to the creative process of our deliverable. We identified how we came up with the “Heat Map” concept by choosing the outline, colours and visual representation of our information. We elaborated how this identified that the type of co-creative work that we engaged in provided many opportunities to build on each other’s knowledge and skills. In comparison, we added an example of when we experienced difficulty in working as a team. This was discussed in the context of having some more vocal students and the tensions that arose during group discussions.
We also clarified what was meant by the community partner about the impact of miscommunication between the CUP in the section on Communication and Miscommunication.
Further development of the insights/recommendations
We have added to the Insights and Recommendations section by providing more concrete examples about what “institutional structure and support” looks like. The following additions were made to this section:
Institutional support must include components such as leadership and support from deans, provosts, and presidents, both for institutional change and for individual faculty (Cavallaro 2016). Further, institutional support means framing faculty work within the context of the institution’s mission which often includes a commitment to “serving society” and assessing the actual work in which faculty are engaged and committed (Calleson, Jordan, and Seifer 2005, 318). This requires changes to tenure and promotion policies that redefine definitions of “scholarship” and recognize and reward CES scholars for a spectrum of scholarship beyond traditional measures of impact such as academic journal articles and books (Calleson, Jordan, and Seifer 2005; Gelmon and Jordan 2013). It also means valuing and rewarding time spent in community partnership formation which is important for building relationships and trust. When possible, we encourage future CES scholars to seek out and use the institutional infrastructure and support that is available within higher education institutions to support principled engagement. For example, this can include seeking out research services assistance in applying for community engaged research and training funding. This can also include working with administrators to support classroom space that is more conducive to the pedagogy (e.g., adequate and accessible meeting space for CU partnerships and for weekly student collaborative group work). It can also mean taking the time to work with librarians, and knowledge mobilization experts to provide specialized knowledge and skills training that neither students nor faculty have, in order to produce scholarship that is more engaging and likely to have greater impact. It can also include working with teaching and learning experts in institutional teaching and learning centres and in particular experts in community engaged learning for support via resources such as rubrics, assessment examples, teaching and learning activities etc. as well as being invited to participate in communities of practice with others who are doing CEL. And if such support is not available, we need to ask for it and encourage institutions to learn from those who are leaders in supporting and rewarding CES.
Address the institutional requirement of grading
We addressed this issue by providing an additional paragraph connecting the issue of power to grading and assessment in the faculty reflection section entitled Trust, Collaboration, and Power Dynamics and the issue of assessment was revisited in the Lessons Learned section at the end.
Reviewer 2 Report
I very much appreciate the kind of experience that all involved had in this project. I have a similar experience every time I do a community-based research (CBR--I use this term because they are interchangeable, see below) project, and I do at least one each year. It is very clear how energizing the project was and how much everyone enjoyed writing this paper. One part of the story I would ask all to reflect upon a bit more is what was actually accomplished through this project. I know it feels good to produce a research report and have a community group use it. I actually did a similar project just last year with students and a community group (only on media portrayals of hip-hop), that the community group used in a similar way. I would not consider the distribution of a research report as an outcome. It is only an output. An outcome would be a change in media coverage policy, that is then reflected in the actual coverage. In our project we have not seen that yet. I would urge all involved to be more reflective on what is required to not just produce knowledge but to make it matter in substantial and documentable ways.
As a story of a CBR project, and reflections upon it from the three major structural actors (nonprofit leader, faculty, students) this is a complete paper. I do not know if that is what the editor is looking for. If it is, I have no critique other than the discussion of outputs versus outcomes that I mentioned. If the editor is looking for something that extends knowledge beyond the current literature, I can offer a few thoughts.
Overall, attempting to limit the literature review to a few people who use specific terms (like CEnR or CEL) creates a fictional separation. Chandler and Torbert ( see their 2003 article, Transforming inquiry and action: Interweaving 27 flavors of action research. Action Research, 1:33–52) identified over two dozen terms for just the research practice fifteen years ago. I'd argue there are three dozen now. Same for CEL, SL, ASL, CBL, and so on for the learning side. Academics like to make a big deal about hair-splitting definitions but that's literally just academic and in practice all the definitions become irrelevant as real-life practices overlap more and more--there is way more variation in actual practice within definitions than between them. Limiting the literature to only people who use a specific arbitrary term severely limits access to the knowledge that applies to the practice, and to this paper.
When one does access all the available knowledge, one sees that the experiences described in this project are quite common and well-known in the literature. There is actually not a general gap as asserted in the paper. The issues identified for each structural player fit the literature quite well. Consequently, based on my knowledge of the literature the paper doesn't appear to question or add to what we already know. One way for the authors to find a way to extend the literature is to gather up the various veins of literature and then focus on only one. There is a vein, for example, on partnership processes. There is a mountain of literature on student reactions and affects. Those two seem the most likely to produce possibilities, but there may be others.
Again, extending the literature may not be important to the editor, and I do not advocate that as a necessity. If telling the story and reflecting upon it is enough, I have no objections. But I know that extending the literature is the standard for most journals, which is why I mention it.
Author Response
Dear Reviewer 2, thank you for your thoughtful feedback. Below is our response to your suggestions/recommendations.
What was actually accomplished through this project?
To answer this question we provided clearer information in the section called Community University Partnership Case Study Overview about what we created.
Accordingly, the main goal of the CES project was to create a knowledge mobilization tool that [provincial not for profit association] could use to inform and educate different audiences (e.g., Ontario women shelter staff, the public, government, journalists, etc.) about how Ontario media sources report and portray the issue of femicide. By the end of the course, we achieved this output. See https://www.oaith.ca/oaith-work/current-initiatives.html for the final knowledge mobilization products co-created by this CU partnership.
We also explain in the section called Meaningful and Applicable Research that our main goal was to create a KMb tool to be used by the community partner. The Heat Map and the aggregate analysis of the media framing analysis were two outcomes we created with the community partner in addition to a methodology and tools for coding and framing. In the section called Lessons Learned we added information about what the outcomes of the project were:
As a community partner, we now have a well-developed process, an evidence-based coding instrument and media framing tool, and meaningful outputs including knowledge mobilization products (i.e., the interactive Heat Map and aggregate analysis) that are both on our website and is an important part of our ongoing education and advocacy about femicide and violence against women.
We also identify outcomes as providing the faculty member with better processes and tools to use in future CEL classes.
We think we have addressed the comment that we should reflect on “what is required to not just produce knowledge but to make it matter in substantial and documentable ways”.
We have clarified the request to distinguish between Outcomes and Outputs in the following way and we thank the reviewer for the recommendation to address this important distinction. We have now clearly identified our knowledge mobilization products as the outputs of our project (e.g., online public femicide visuals), while acknowledging that the ultimate intended long-term outcome of our outputs is to promote gender equality and social justice.
We have added the following points to elaborate on this distinction in the section called Community University Partnership Case Study Overview:
Although it is still too early to assess the full impact of our knowledge mobilization outputs, our community partner has provided us with feedback and metrics that help to exemplify the immediate outcomes of our work thus far. Specifically, our online femicide visualization has had over 1700 views which continue to increase steadily every year, indicating substantial public reach. Moreover, since its online publication, our partner has been using our femicide research and knowledge mobilization tool to present evidence-based information in conversations with their funders. Our partner has credited our research and tool for helping them present a stronger case for gaining increased funding for femicide resources, training, and public awareness, resulting in significant increased funding in recent years.
Extending and broadening the nomenclature and the literature
To address the reviewer’s comment about more clearly discussing how our work is defined, we added a further explanation about why we use the term “community engaged learning” (CEL). Specifically, the reviewer commented on the nomenclature that is used to refer to various versions of Action Research. We have broaden the connections to other nomenclature and broadened theliterature and we make a stronger argument about why we think this distinction is important and what CEL entails, and cited scholars who use this type of research (including Boland, 2013; Bringle and Hatcher, 1995; Gordon da Cruz, 2017; Kleinhesselink, 2015; Johnson-Curiskis, and Wolter, 2004; Morton, 2013; Weigert, 1998). Furthermore, we included more information about why we wanted to increase interest in and understanding of the community engagement principles and practices to encourage more scholars to apply to this process to their work. We also emphasized the lack of literature on CEL specifically and how this adds to making our work unique.
With regards to the teaching-research-service, we identified that this nexus is not new and that many scholars are increasingly identifying different ways of addressing this continuum. We further clarify the contribution of our article throughout the paper by explaining how we are advancing the teaching-research-service nexus with a focus on CES and CEL. Specifically, we describe that CEL and CES has the potential to unite and integrate the teaching-research-service components in mutually beneficial ways within a community university partnership that are less emphasized with other approaches (e.g., service learning).
Round 2
Reviewer 1 Report
Thank you so much to the authors for responding to the comments and suggestions. I really appreciated the thoughtful nature of the responses and the contribution to the field.